# Investigation of the Thermodynamic Transformation and Rare Earth Microalloying of a Medium Carbon-Medium Alloy Steel for Large Ball Mill Liners

**DOI:** 10.3390/ma15134448

**Published:** 2022-06-24

**Authors:** Xi Yang, Xinghe Wang, Jingpei Xie, Yuhe Li, Dixin Yang

**Affiliations:** 1Guangdong Provincial Research Center for Sports Assistive Device Design Engineering and Technology, Guangzhou Sport University, Guangzhou 510500, China; yangxi_anne@hotmail.com (X.Y.); 13922738963@139.com (Y.L.); 2School of Materials Science and Engineering, Henan University of Science and Technology, Luoyang 471000, China; wangxinghe6@126.com

**Keywords:** medium carbon-medium alloy steel, continuous cooling transition curve, heat treatment, multivariate microalloying, microstructure

## Abstract

The medium carbon-medium alloy steel was developed for the manufacture of large ball mill liners and sports equipment. In this study, the continuous cooling transformation curve of a novel type of medium carbon-medium alloy steel was measured with a thermal simulation machine; based on this curve, the hardening and tempering processes were optimized. The steel was then complex modified with alkaline earth and rare earth alloys. The mechanical properties of the treated steel were tested. The microstructure of the steel was analyzed by metallographic microscopy, X-ray diffraction, scanning electron microscopy and transmission electron microscopy, and the wear surface of the steel was analyzed by a three-dimensional morphometer. After high-temperature tempering, the microstructure transformed into tempered sorbite, which possesses good mechanical properties and can adapt to working conditions that require high strength and toughness. Rare earth or alkaline earth modification of the medium carbon-medium alloy steel promoted microstructural uniformity and grain refinement and improved the mechanical and anti-wear properties.

## 1. Introduction

With the increasing use of large mills in mining, cement, and other industries, the requirements for the quality of mill liners have changed. Materials with higher strengths and better toughness and wear resistance properties are needed to resist abrasion or drilling wear under strong-impact and high-stress working conditions. Traditional materials such as high manganese steel cannot meet these new demands because of their low yield strengths and distortion under strong-impact working conditions. XW42 steel, bimetal composite plates, cast nickel-base hard alloy IV, and clad plates of chromium steel and silicon carbide have been proven to be ideal materials for liner plates [1,2], but their complicated manufacturing processes and high costs limit their use in large ball mills. Medium carbon steel is one of the most widely used ferrous metals on the market today. By using a proper heat treatment process, the microstructure and mechanical properties of medium carbon steel can be adjusted to adapt to the high standards required in transportation, aerospace, space, underwater, and other working conditions [3]. The current trend is to use medium/high carbon alloy steel to produce liner plates for large mills.

The function of the lining plate is to protect the barrel body from direct impact and friction exerted by the grinding media and materials. Liner failure often results from both internal and external factors. Technological optimization of casting and heat treatment, alloy composition design and microalloying are the main methods for improving the anti-friction, anti-deformation, and anti-fracture characteristics of medium carbon steel [4,5,6]. Extensive studies and diverse processing approaches have been carried out to alter the microstructure of alloy steel and improve its impact resistance, stiffness, wear resistance, and strength [7,8,9,10,11,12]. Abbasi et al. analyzed the microstructural features of three different types of medium carbon steel (namely, MnCrB-, NiCrSi-, and NiCrMoV-bearing steels) and found that the addition of small amounts of alloying elements and varying austenitic grain sizes were mainly responsible for the changes in the microstructure and mechanical properties of the steel [13]. Liu et al. studied the effects of heating temperature (850–1100 °C) and heating time (30–150 min) on the growth behavior of austenite grains in a medium carbon alloy steel and established a predictive model to describe the growth of austenite grains during the heating process [14]. Kadowaki et al. investigated the optimal tempering conditions for a martensitic medium carbon steel (mass: 0.47% C) to balance ductility and pitting resistance [15]. Shi et al. studied the high-temperature sliding wear behavior of a promising high-boron medium carbon alloy (HBMCA) and found that the mechanism underlying high-temperature wear of the HBMCA included oxidation wear, abrasive wear, and adhesion wear [16]. Rare earth modification of steels has been highly studied in steel metallurgy [17,18,19,20]. The study of rare earth elements (Re) in low carbon steels shows that Re inhibit the phase transformation of ferrite. The grain structure of bainite can be improved by rare earth metal microalloying [17]. The addition of the Re, La, and Ce (200 ppm) to low-carbon cast steel can change the interaction (including the peritectic transformation) between inclusions and the matrix, thus affecting the microstructure and impact properties of the alloy [18]. The continuous cooling transformation (CCT) curve is essential in determining the heat treatment process [19,20]. Jung et al. improved the transformation model describing transformation strain as a function of phase fraction through both experimental and numerical approaches and obtained more accurate CCT kinetics for the simulation of the quenching process for plain medium carbon steel [20].

ZG_70_Cr_2_MnNiSi steel is a medium carbon-medium alloy steel that was developed in recent years. The total alloy element content in this steel is 6–8%, and low amounts of valuable rare elements are present. Therefore, this steel is relatively inexpensive and easy to popularize and apply. The relationship between the strength and toughness of this alloyed steel can be controlled over a wide range by complex alloying. ZG_70_Cr_2_MnNiSi steel exhibits the characteristics of wear-resistant steel, which is applicable in the production of lining boards for large ball mills and is also a suitable material for critical components in fitness equipment that require a high bearing capacity [3]. This study focuses on improving the production process, microstructure, and properties of ZG_70_Cr_2_MnNiSi steel.

## 2. Materials and Methods

### 2.1. Composition Design

To compare the effect of modification on the microstructure and properties of medium carbon alloy steel, three different processes were adopted in the preparation of the tested steels: no modification agent was added to the ladle, only the Sr-Si-Fe alkaline earth metal composite was added, and only the Re-Ca-Ba rare earth metal composite was added. The compositions of the tested steels prepared by three different processes are shown in Table 1.

### 2.2. Specimen Preparation

The tested steels were obtained by melting in a medium-frequency induction furnace. First, all raw materials were prepared according to the alloy composition design. Pig iron, scrap steel, nickel, high-chromium cast iron, ferromanganese, ferrosilicon, and ferromolybdenum were added consecutively to the induction furnace and then heated by ohmic heating. When the material in the furnace became molten, rare earth metal pieces were placed at the bottom of a ladle, and the amount of rare earth metal added is provided in Table 1. When the temperature of the molten steel in the furnace reached 1550–1580 °C, the melt was poured into the ladle, and, through this shock melting step, the rare earth metal was dissolved in the liquid steel to act as a microalloying modification agent. Thereafter, the liquid steel was poured into a mold made from sodium silicate sand and cast into a liner (Figure 1). The ball mill liner was transported to the corresponding position on the inside surface of the ball mill barrel by mechanical lifting equipment, and the liner was locked on the barrel body with bolts. After a series of liners were installed, the wear working surface of the ball mill was formed. The specimens were cut from the liner by wire cutting and processed into different sizes for CCT curve measurements and heat treatment or mechanical property tests and microstructure observations. The sizes of the specimens were Φ6 × 10 mm for the phase transformation point tests, 12 × 12 × 15 mm for metallographic observations, Φ10 × 90 mm for tensile tests, and 12 × 12 × 15 mm for determination of the constituent phases by XRD.

### 2.3. Experimental Methods

First, we determined the CCT curve of the steel and then optimized the quenching and tempering processes. A rare earth alloy was also added to modify the steel. The effects of the heat treatment process and rare earth microalloying on the microstructure, mechanical properties, and wear resistance of the tested steels were examined by X-ray diffraction (XRD), Bruker Ltd., Karlsruhe, Germany, scanning electron microscopy (SEM), JEOL Ltd., Kyoto, Japan, transmission electron microscopy (TEM), Hitachi Ltd., Tokyo, Japan and a universal material testing machine, Shimadzu Ltd., Kyoto, Japan.

The CCT curve of the ZG_70_Cr_2_MnNiSi steel was measured by a Gleeble-1500 thermal simulation machine (Data Sciences International, Inc., Saint Paul, MN, USA). The alloy specimens were first heated to 950 °C at a heating rate of 10 °C/s and then held at this temperature for 5 min, during which Ac_1_, Ac_3,_ and Ms were measured. They were then cooled from Ac_3_ to room temperature at rates of 0.05, 0.1, 0.15, 0.2, 0.5, 1, 4, 10, 15, 30, 40, and 45 °C/s. The phase transition points for the different cooling rates were determined by applying the tangent method to the thermal expansion curve. The CCT curves were constructed by connecting the phase transition start and end points of the same phase with curves.

According to the characteristics of medium carbon alloy steel, the heat treatment process was devised as follows. First, the samples were annealed. The specimens were heated to 1050 °C, held at this temperature for 2 h, cooled to 500 °C together with the furnace, and subsequently removed from the furnace and cooled in air to room temperature. Second, a quenching treatment was performed. The annealed specimens were reheated to 860 °C and held for 2 h. They were then removed from the furnace and quenched in oil to room temperature. The final step was tempering. The quenched specimens were heated to 550, 570, 590, and 610 °C and held at each temperature for 2 h before being removed from the furnace and cooled in air to room temperature. The heat treatment equipment used in our study included a VFX9/160-YG high-temperature resistance heating box furnace (Shanghai Yifeng Electric Furnace Co., Ltd., Shanghai, China) and an SJX-4-13 medium-temperature resistance heating box furnace (Shanghai Yifeng Electric Furnace Co., Ltd., Shanghai, China).

The mechanical properties of the steel under static tensile loading were determined by a Shimadzu AG-250KN precision universal material testing machine (Shimadzu Corporation, Kyoto, Japan). Rockwell hardness was measured under a load of 1500 kg by an HRD 150 electric Rockwell hardness tester (Laizhou Lailuote Testing Instrument Co., Ltd., Laizhou, China). Vickers hardness was measured under a load of 300 g by an MH-3 microhardness tester (Minsks Test Equipment Co., Ltd., Xi’an, China). Due to the small size of the specimens for Vickers hardness testing, samples were embedded in a metallographic resin prior to being tested. The friction and wear properties were tested by an ML-100 abrasive wear machine (Jinan Yihua Tribology Testing Technology Co., Ltd., Jinan, China). The specimen size was Φ6 × 20 mm, the load was set to 20 N, and the feed radius was 120 cm. The rotary speed of the abrasive wear testing machine disk was set at 60 r/min, and #240 sandpaper was used. We measured the mass of the specimens before and after wear using an analytical balance and obtained the average value of three specimens. The wear surface morphology of the specimens was observed by a NanofocusAG 3D morphology tester (Nanofocus Corporation, Frankfurt, Germany).

The metallographic structure of steel was observed by an OLYMPUS PMG3 optical metallographic microscope (Olympus Optical Co., Ltd., Tokyo, Japan) and a JSM-5610LV scanning electron microscope (JEOL Ltd., Kyoto, Japan). Phase analysis was performed using a D8ADVANCE X-ray diffractometer (Bruker Corporation, Karlsruhe, Germany). The fine structure of the alloy was observed using an H-800-1 transmission electron microscope (Hitachi Ltd., Tokyo, Japan), and the morphology, quantity, and distribution of carbides were observed by selected area electron diffraction.

## 3. Results

### 3.1. CCT Curve and Phase Transformation of ZG_70_Cr_2_MnNiSi Steel

The microstructure and mechanical properties of medium/high carbon alloy steels are significantly influenced by their quenching and tempering temperatures, and phase transition kinetics control the resulting metallographic structure. The ZG_70_Cr_2_MnNiSi steel in this study is a hypoeutectoid steel (C < 0.77%), and hypoeutectoid steel has a bainite transition zone. The critical temperatures of the ZG_70_Cr_2_MnNiSi steel were measured as Ac_1_ = 736 °C, Ac_3_ = 811 °C, and Ms = 218 °C. The CCT curve of the ZG_70_Cr_2_MnNiSi steel is shown in Figure 2. Transformations ranged from austenite to ferrite and pearlite (A → F and P), austenite to bainite (A → B), and austenite to martensite (A → M) depending on the cooling rate. The pearlite and bainite transformation zones of the tested steels are shown on the right in Figure 2.

The microhardness curves of the tested steels changed with the cooling rate, as shown in Figure 3. The microhardness gradually increased with increasing cooling rate. The microhardness increased sharply when the cooling rate increased from 0.05 °C/s to 0.15 °C/s. When the cooling rate increased from 15 °C/s to 45 °C/s, the microhardness did not change significantly. Under these conditions, the structure of the steel consisted of a mixture of martensite and a very small amount of residual austenite, and the hardness of martensite tended to be stable.

The metallographic structure of the ZG_70_Cr_2_MnNiSi steel at different cooling rates is shown in Figure 4. When the cooling rate was 0.1 °C/s, the phase transition products were ferrite, pearlite, bainite, and a small amount of residual austenite. When the cooling rate was 0.2 °C/s, the phase transition products were a mixture of a small amount of acicular martensite and residual austenite. When the cooling rate was between 0.5 and 45 °C/s, the phase transition product was mainly martensite.

### 3.2. Effects of Heat Treatment on the Microstructure and Properties of ZG_70_Cr_2_MnNiSi Steel

The heat treatment process for lining boards usually consists of quenching and high-temperature tempering. Martensite is formed during quenching and increases the hardness of the lining plate. Tempering after quenching can reduce the internal stresses in the steel, stabilize its microstructure, and increase its plasticity. Therefore, the rational design of quenching and tempering processes can significantly improve the toughness of steel.

A variety of alloying elements, such as silicon, manganese, chromium, nickel, and molybdenum, were added to the alloy steel in our experiment. The microstructure of the steel was bulky and contained many precipitated dendritic crystals. First, the cast grains were refined and homogenized by high-temperature annealing to prepare the desired microstructure for subsequent quenching.

#### 3.2.1. Influence of Tempering Temperature on the Hardness and Strength of ZG_70_Cr_2_MnNiSi Steel

The relationship between the tempering temperature and hardness of the ZG_70_Cr_2_MnNiSi steel is shown in Figure 5. With increasing the tempering temperature, the steel hardness decreased due to the decomposition of martensite and residual austenite and the transformation of carbides. In the range of 570 °C~590 °C, the hardness decreasing trend becomes weak and secondary hardening occurs. After tempering at 590 °C, the hardness of the tested steel reached approximately 45 HRC, which meets the wear resistance requirements for lining plates.

In Figure 6, the tensile strength of the experimental steel decreased gradually with increasing tempering temperature; however, the amplitude was small. The yield strength declined more sharply. Figure 7 shows that both the area and elongation decreased significantly with the increasing tempering temperature.

#### 3.2.2. Metallographic Structure of ZG_70_Cr_2_MnNiSi Steel after Tempering

The microstructure of the tested steel after quenching from 860 °C was comprised of lath martensite and residual austenite. The microstructure of tempered steel was composed of tempered sorbite, residual austenite, and granular carbide. The tempered sorbite has excellent strength and plasticity. At 550 °C~590 °C, the amount of Mo_2_C and other fine carbide precipitates increases with the increase in tempering temperature, which is advantageous to the refinement of the microstructure. However, when the temperature rises to 610 °C, the phenomenon of carbides growing up appears (Figure 8).

#### 3.2.3. Influence of Tempering Temperature on the Abrasive Wear Performance of ZG_70_Cr_2_MnNiSi Steel

Abrasive wear test results are shown in Table 2. The amount of wear changed little in the tempering temperature range of 550 °C~570 °C. When the tempering temperature exceeded 590 °C, the amount of wear increased significantly. The three-dimensional morphology of the grinding surface of the specimen after tempering at 590 °C is shown in Figure 9. Obvious furrows and extrusion edges are observed on the worn surface, and the furrows are evenly distributed.

The quenching temperature of medium carbon steel is usually set to be 30–70 °C above the actual temperature of phase change Ac_3_, and Figure 2 shows that Ac_3_ is 811 °C, so the quenching temperature was determined to be 860 °C. As for the next step of high-temperature tempering, and a total of four groups of tempering temperatures of 550, 570, 590, and 610 °C were formulated in this study, and the best tempering temperature of 590 °C was selected by testing the performance of each group. Consequently, the heat treatment process based on the above experiments is designed as follows. The experimental steel samples were heated to 860 °C for 2 h and then quenched by spray cooling. The quenched steel was held at 590 °C for 2 h and then air cooled for tempering treatment. The hardness of the experimental steel after the quenching and tempering treatment was 44.83 HRC, the tensile strength (Rm) was 1285 MPa, the lower yield strength (ReL) was 1188 MPa, the reduction in the section rate (Z) was 4.68%, and the elongation after fracture rate (A) was 4.26%. The steel was endowed with a proper combination of hardness and toughness, and can meet the requirements of both impact and wear resistance.

### 3.3. Rare Earth Compound Modification of ZG_70_Cr_2_MnNiSi Steel

#### 3.3.1. Influence of Rare Earth Compound Modification on the Mechanical Properties of ZG_70_Cr_2_MnNiSi Steel

The hardness results for the steel specimens with and without rare earth modification are shown in Table 3. It should be mentioned that the three types of steel specimens were processed by the same optimized heat treatment procedure (quenching at 860 °C and tempering at 590 °C). The addition of either Sr-Si-Fe or Re-Ca-Ba compound modifiers had no obvious effect on the hardness of the steel.

Tensile test results are shown in Table 4. Rm of Sr-Si-Fe-treated steel increased by 58 MPa, ReL increased by 38 MPa, and the elongation after fracture A, and the percent section shrinkage Z increased to different extents. After Re-Ca-Ba modification, Rm, ReL, the elongation after fracture, and percent section shrinkage all increased; however, overall, the magnitudes of these increases were not as large as those for steel modified by Sr-Si-Fe.

#### 3.3.2. Influence of Rare Earth Compound Modification on the Phase Composition of ZG_70_Cr_2_MnNiSi Steel

The XRD analyzer utilizes Cu-Kα radiation with a wavelength of 1.54 Å. It can be seen from Figure 10 that, due to the addition of various alloying elements in steel, it is easy to form multiple carbides. M in M2C mainly refers to Mo. M3C mainly refers to (Fe, Mo, Cr, Mn)3C, and M7C3 refers to (Cr, Fe)7C3. The effect of rare earth compound modification on the tested steels is reflected mainly in the intensity of the diffraction peaks. The intensities of the α-Fe, Fe3C, M7C3, and M2C diffraction peaks in the XRD patterns of specimens #2 and #3 were higher than those in specimen #1. The increase in the diffraction peak intensity of α-Fe is related to the orientation of the crystal plane, and the increase in the intensity of the carbide diffraction peaks indicates that rare earth modification is beneficial to the formation of fine carbides.

#### 3.3.3. Influence of Rare Earth Compound Modification on the Metallographic Structure of ZG_70_Cr_2_MnNiSi Steel

The metallographic structure of the tempered steel is shown in Figure 11. High-temperature tempering resulted in the formation of a mixture of ferrite and granular carbides. The acicular morphology of α-martensite, also referred to as high-temperature tempered martensite, did not disappear completely. A comparison of Figure 11a–c shows that the re-modified samples in Figure 11b,c have smaller particle sizes and more uniform distributions than the sample in Figure 11a, indicating that rare earth modification plays a positive role in grain refinement and microstructural homogenization.

The test steel was treated with both Sr-Si-Fe and Re-Ca-Ba composite, and similar results were achieved in terms of metallographic organization, mechanical properties, and anti-wear properties. The observation of the fine structure of the test steel by transmission electron microscopy requires a large amount of preparation work. Considering that there has been lots of research about rare earth metamorphic steels, but few on alkaline earth metamorphic steels, here we only analyzed Sr-Si-Fe metamorphic treated ZG_70_Cr_2_MnNiSi steel in the TEM test, as shown in Figure 12. The diffraction pattern in Figure 12a indicates the presence of high-temperature tempered martensite. As shown in Figure 12b, bright white films are present between the martensite laths; these films were characterized, by selected area electron diffraction, as residual austenite. Modification treatment can promote the formation of high-density dislocations and facilitate the precipitation of a considerable amount of carbides during tempering. Fine carbides were distributed in the lath martensite grains, as shown in Figure 12c,d. The carbide shapes included long rods and round, elliptical and spherical particles; among these, spherical carbides were the most common.

#### 3.3.4. Influence of Rare Earth Compound Modification on the Wear Resistance of ZG_70_Cr_2_MnNiSi Steel

The surface morphology of the tested steels after abrasive wear is shown in Figure 13. Furrows and extrusion edges are distributed over the abraded surface. In Figure 13a, microcracks appear on the subsurface of the grinding surface, and the growth of these microcracks leads to spalling of the grinding surface. In Figure 13b,c, the extrusion edges are uniformly distributed. The loss of the tested steels with and without modification by abrasion is shown in Table 5. The relative abrasion losses of rare-earth-modified specimens #2 and #3 are the same, and both show values that are approximately 10% lower than that of the unmodified specimen #1.

## 4. Discussion

### 4.1. Composition Characteristics and CCT Curve of ZG_70_Cr_2_MnNiSi Steel

ZG_70_Cr_2_MnNiSi steel is a medium carbon-medium alloy steel that was developed in recent years. Its carbon content not only ensures that a sufficient amount of carbides is present in the structure but also prevents brittle fracture and difficult processing caused by an excessive carbon content. The reduction in the amounts of Mo, Ni, and other valuable rare metals can reduce the cost. By adjusting the proportions of common alloying elements (Si, Mn, and Cr), adopting rare earth compound modification, and optimizing the heat treatment process, we ensure that this steel has sufficient mechanical performance, in terms of both strength and toughness, to meet the manufacturing requirements for the lining plates of large ball mills.

As seen from the CCT curve of the ZG_70_Cr_2_MnNiSi steel, pearlite and bainitic transition zones are present at the right because all alloying elements, except Co, dissolved in the austenite phase. The Ms point of the steel is 218 °C, and the right half of the Ms line is reduced, which is caused by the precipitation of proeutectoid ferrite and the transformation of bainite, leading to the surrounding austenite being rich in carbon. When the tested steel was cooled at a low cooling rate, pearlite was formed in the high-temperature region. When the temperature dropped to 400 °C, supercooled austenite decomposed into bainite by the half diffusion mode phase transition. Bainite precipitated preferentially along the grain boundaries. When the temperature was further decreased, neither iron nor carbon atoms were able to diffuse, and only crystal lattice rearrangement and martensitic transformation occurred. With the increasing cooling rate, the temperature at which supercooled austenite transformed into ferrite, pearlite, and bainite decreased gradually, and the quantity of the three products decreased as well; however, the amount of martensite formed increased. In this study, when the cooling rate was greater than 0.2 °C/s, only martensitic transformation occurred, indicating that the ZG_70_Cr_2_MnNiSi steel has good austenitic stability. Even at slow cooling rates, only the martensitic structure was obtained. Hence, inexpensive air cooling can be adopted to prepare martensitic wear-resistant steel matrices in actual production.

### 4.2. Influence of the Tempering Process on the Microstructure and Properties of ZG_70_Cr_2_MnNiSi Steel

The tested steel contains carbide-forming elements. For example, Cr can form alloy cementite (Fe, Cr)3C. The addition of Mo in the steel will induce the replacement of Fe and Cr with Mo in (Fe, Cr)3C. As the tempering temperature increased from 550 to 590 °C, the alloy carbides gradually precipitate and extremely fine alloy carbide particles were distributed homogeneously in the ferrite matrix. Due to the high affinity between carbon and the alloy elements in alloy cementite, fine grains of alloy cementite are difficult to dissolve, hindering the aggregation and growth of cementite [21]. However, when the tempering temperature exceeds 590 °C, the hardness, yield strength, and wear resistance of steel decrease obviously due to the aggregation and growth of cementite under high temperature.

### 4.3. Influence of Rare Earth Compound Modification on the Microstructure and Properties of ZG_70_Cr_2_MnNiSi Steel

The ZG_70_Cr_2_MnNiSi steel treated by rare earth compound modification possessed a homogeneous microstructure, refined grains, and improved strength and plasticity. The TEM analysis and electron diffraction patterns revealed that the high-temperature tempered structure of the steel consisted of lath martensite and retained austenite at the lath boundaries. Rare earth compound modification promoted the formation of a high-density martensite dislocation substructure and inhibited the formation of a martensite twin substructure. Under tempering conditions, carbides precipitated along the twin boundaries. The existence of a martensite twin substructure will prevent slipping and lead to increased fragility, whereas the dislocation substructure allows for microdislocation mobility, which can reduce the local stress concentration and increase toughness. The formation of high-density dislocations is also advantageous to the precipitation of a large amount of uniformly dispersed fine carbides within the lath martensite grains. All of these factors are beneficial in enhancing steel hardness and toughness.

## 5. Conclusions

The transformation point tests showed that the martensitic transformation point (Ms) of the ZG_70_Cr_2_MnNiSi steel was 218 °C. The CCT curve showed that the critical cooling rate for the martensite transition was 0.2–0.5 °C/s; within a cooling rate range of 0.5–45 °C/s, martensite and residual austenite were obtained. The critical point temperatures determined in our study were Ac_1_ = 736 °C and Ac_3_ = 811 °C and these can provide a baseline for the quenching and tempering processes of martensitic steel.

A sorbite structure was obtained after the high-temperature tempering of ZG_70_Cr_2_MnNiSi steel. The optimal heat treatment process of the medium carbon-medium alloy steel is summarized as follows: heat treatment at 860 °C for 2 h, quenching, heat treatment at 590 °C for 2 h, and tempering. The hardness index (HRC) of the steel was 44.83, Rm was 1285 MPa, and ReL was 1188 MPa; the tested steel is adaptable to working conditions in which high hardness and toughness of materials are required.

TEM observations showed that the addition of Sr-Si-Fe and Re-Ca-Ba alloys to tested steel is advantageous to the formation of high-density dislocations in martensite. It not only relieves the local stress caused by the concentration of crystal substructure, but also facilitates the diffusion and precipitation of carbides in lath martensite during tempering, thus enhancing the strength and toughness of steel. The relative wear loss was reduced by 10% compared with the steel without rare earth modification.

## Figures and Tables

**Figure 1 materials-15-04448-f001:**
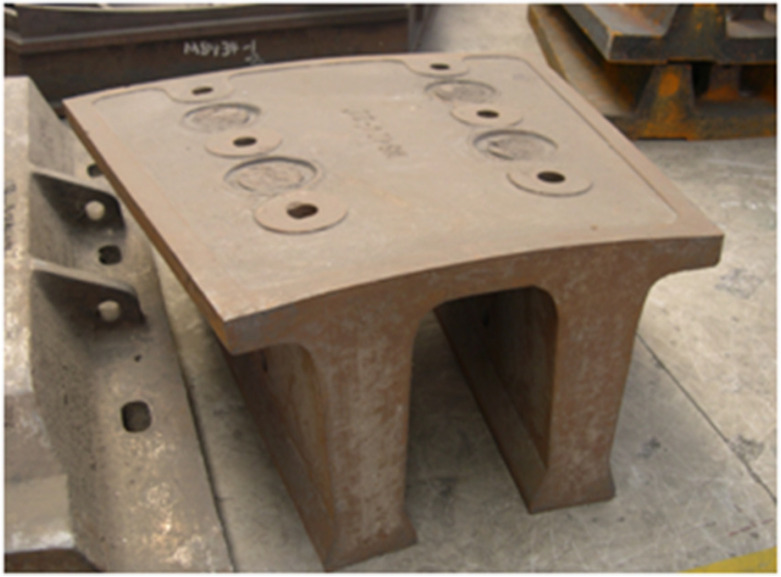
The liner of a large ball mill that was commercially produced with ZG_70_Cr_2_MnNiSi steel by CITIC Heavy Industries Co., Ltd., Luoyang, China. It is mainly used in Φ5500 × 8500 and other large ball mills.

**Figure 2 materials-15-04448-f002:**
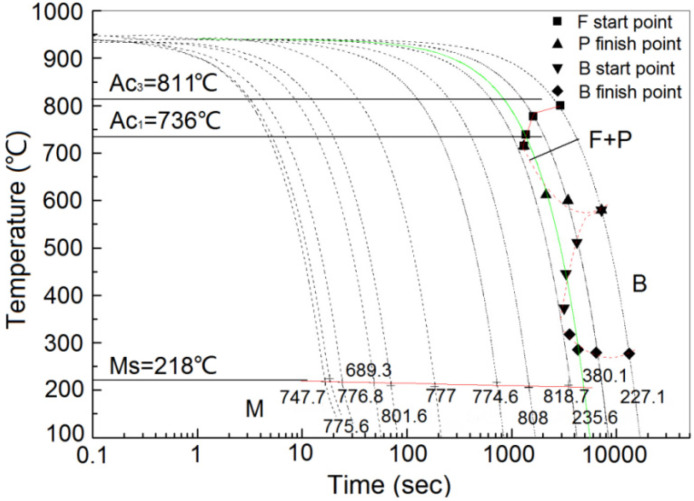
CCT curve of ZG_70_Cr_2_MnNiSi steel. Ms represents the starting point for the martensite transition, A represents supercooled austenite, F and P represent the ferrite and pearlite transition zones, respectively, B represents the bainite transition zone, and M represents the martensite transition zone. The numbers given at the bottom of each cooling curve represent the average hardness values at room temperature. The cooling rates of the curves from right to left are 0.05, 0.1, 0.15, 0.2, 0.5, 1, 4, 10, 15, 30, 40, and 45 °C/s, respectively.

**Figure 3 materials-15-04448-f003:**
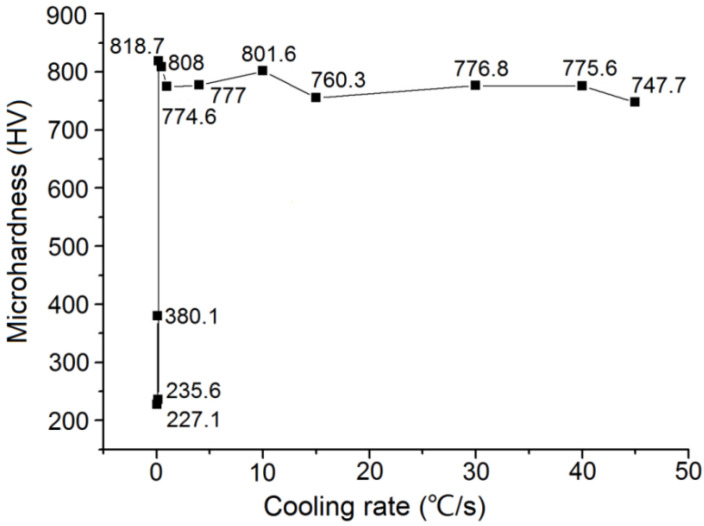
The curve of the microhardness of ZG_70_Cr_2_MnNiSi steel changing with the cooling rates.

**Figure 4 materials-15-04448-f004:**
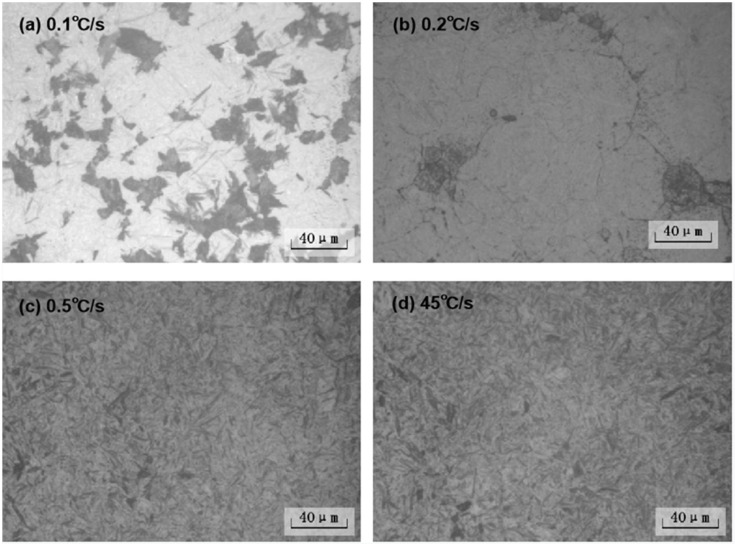
Microstructures of tested steels under different cooling rates.

**Figure 5 materials-15-04448-f005:**
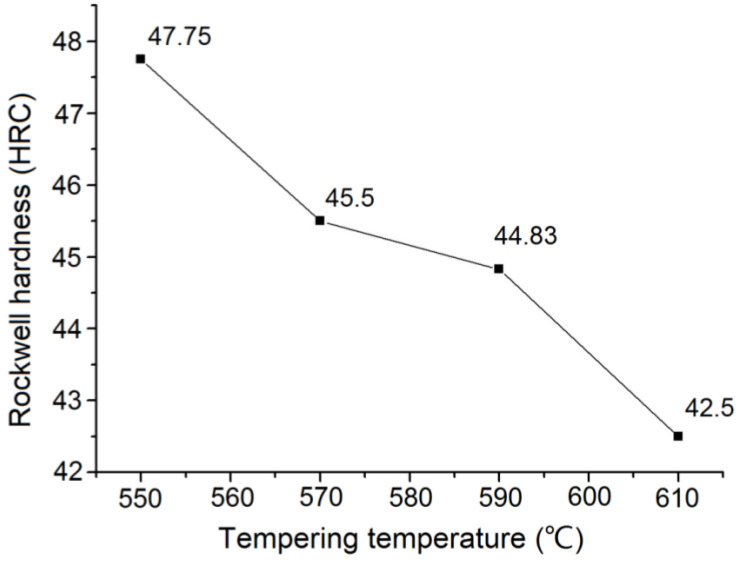
Curve of hardness of tested steels changing with tempering temperature.

**Figure 6 materials-15-04448-f006:**
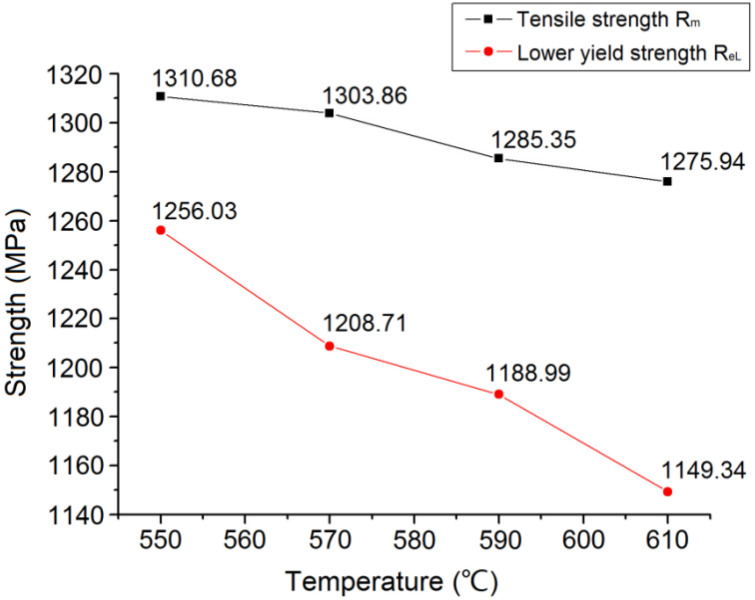
Curve of strength of tested steels changing with tempering temperature.

**Figure 7 materials-15-04448-f007:**
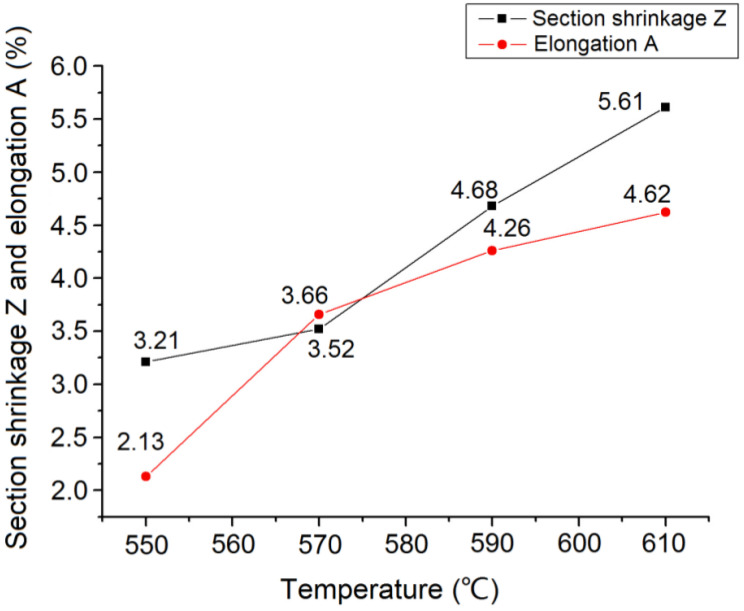
Curve of plasticity of tested steels changing with tempering temperature.

**Figure 8 materials-15-04448-f008:**
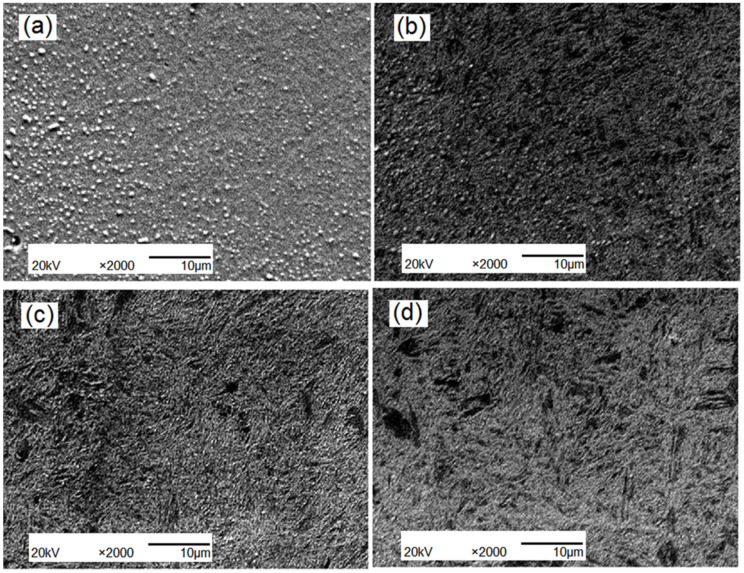
The microstructure of specimens under different temper temperatures: (**a**) 550 °C; (**b**) 570 °C; (**c**) 590 °C; (**d**) 610 °C.

**Figure 9 materials-15-04448-f009:**
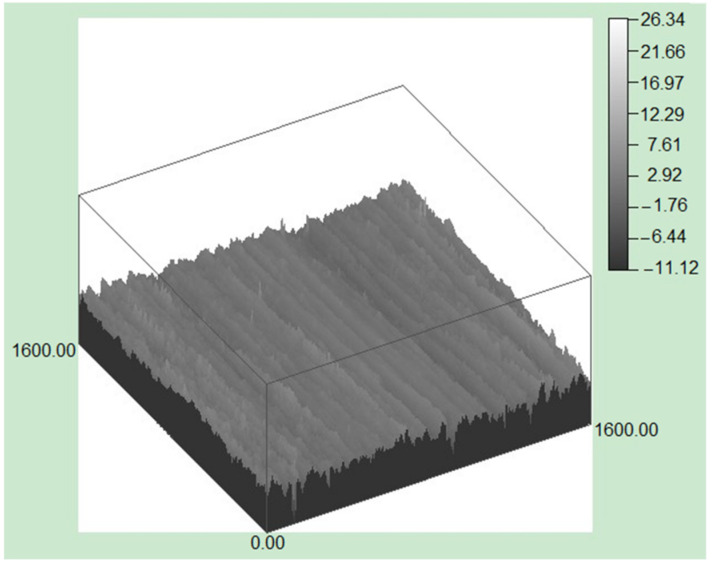
The three-dimensional morphology of the grinding surface of tested steels.

**Figure 10 materials-15-04448-f010:**
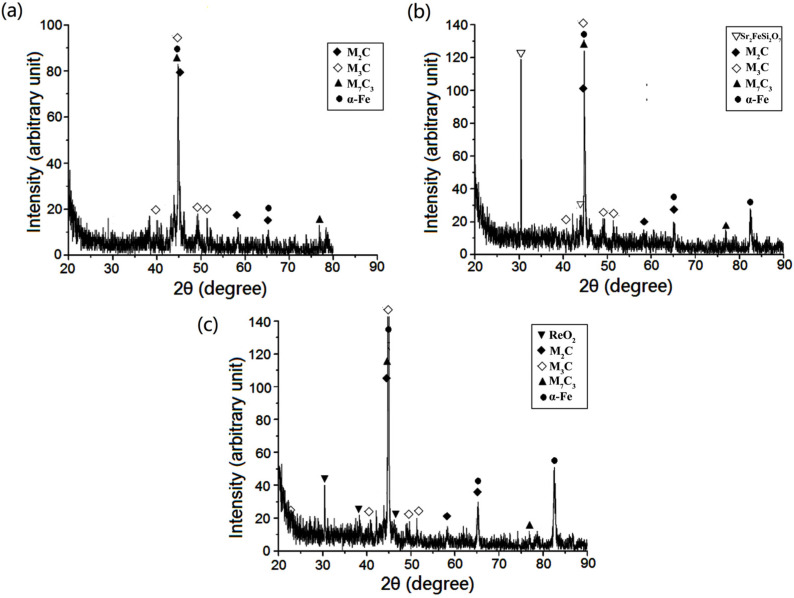
Phase diffraction profile of tested steels: (**a**) #1: ZG_70_Cr_2_MnNiSi steel; (**b**) #2: Sr-Si-Fe modification; (**c**) #3: Re-Ca-Ba modification.

**Figure 11 materials-15-04448-f011:**
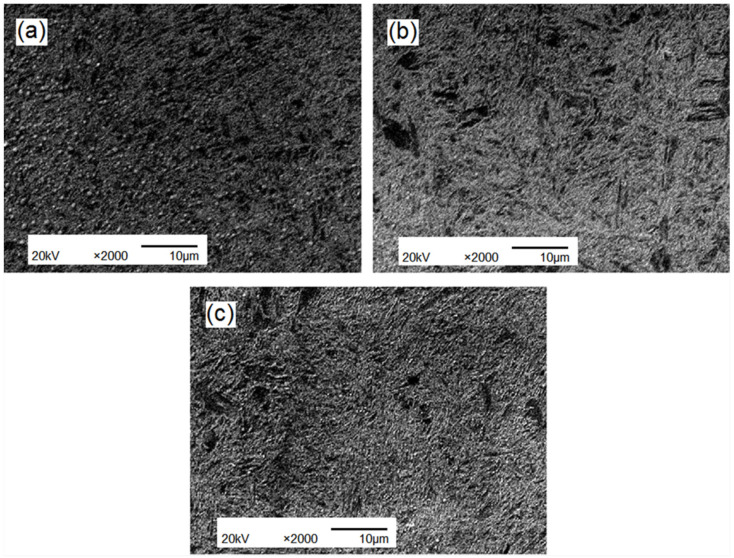
The metallographic structure of tested steels after tempering: (**a**) ZG_70_Cr_2_MnNiSi steel; (**b**) Sr-Si-Fe modification; (**c**) Re-Ca-Ba modification.

**Figure 12 materials-15-04448-f012:**
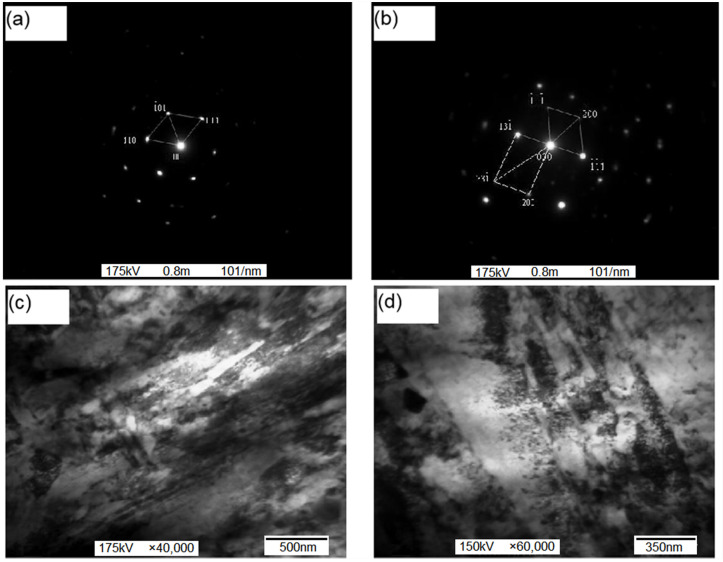
The TEM photos of tested steels: (**a**) martensite; (**b**) residual austenite; (**c**) high density dislocation; (**d**) carbide.

**Figure 13 materials-15-04448-f013:**
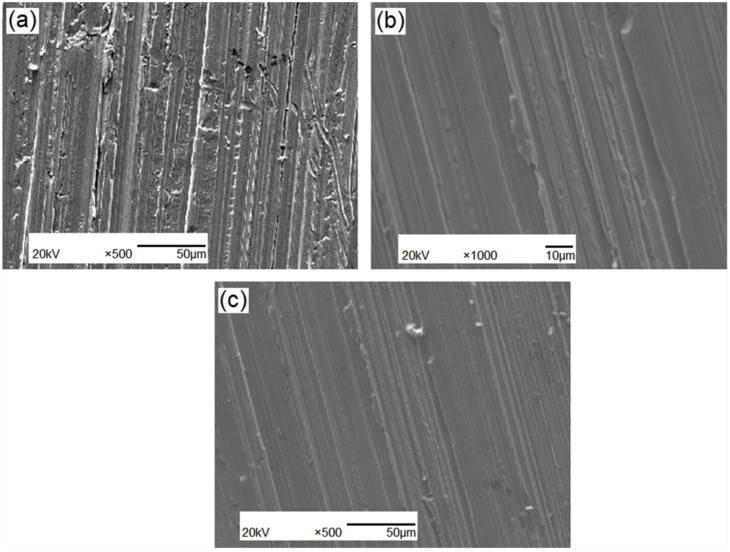
Wear surface morphology of tested steels: (**a**) 70Cr2MnNiSi steel; (**b**) Sr-Si-Fe modification; (**c**) Re-Ca-Ba modification.

**Table 1 materials-15-04448-t001:** Chemical composition of tested steels (wt %).

Tested Steels	C	Mn	Si	Cr	Ni	Mo	Sr-Si-Fe	Re-Ca-Ba
ZG_70_Cr_2_MnNiSi steel	0.6–0.7	0.8–1.2	0.7–1.0	1.5–2.0	0.75–1.0	0.3–0.4		
Sr-Si-Fe modification	0.6–0.7	0.8–1.2	0.7–1.0	1.5–2.0	0.75–1.0	0.3–0.4	0.18	
Re-Ca-Ba modification	0.6–0.7	0.8–1.2	0.7–1.0	1.5–2.0	0.75–1.0	0.3–0.4		0.1

**Table 2 materials-15-04448-t002:** The wear mass loss of steel specimens tempered at different temperatures in the abrasive wear test.

**Tempering temperature**	550	570	590	610
**Average wear (g)**	0.0676	0.0681	0.0748	0.0793

**Table 3 materials-15-04448-t003:** The hardness of ZG_70_Cr_2_MnNiSi steel and the steel specimens modified by two rare earth modification agents.

Steel Specimen	ZG_70_Cr_2_MnNiSi Steel	Sr-Si-Fe Modification	Re-Ca-Ba Modification
**HRC**	42.50	44.83	44.00

**Table 4 materials-15-04448-t004:** Effect of rare earth complex metamorphism on strength and plasticity of test steel.

Steel Specimen	ZG_70_Cr_2_MnNiSi Steel	Sr-Si-Fe Modification	Re-Ca-Ba Modification
**Tensile strength Rm (MPa)**	1227.23	1285.35	1303.86
**Lower yield strength ReL (MPa)**	1170.64	1208.71	1188.99
**Reduction of area Z (%)**	4.68	5.49	5.23
**Elongation A (%)**	4.26	4.87	4.47

**Table 5 materials-15-04448-t005:** Abrasion loss of tested steels.

Specimen Number	Before the Wear Test M0 (g)	After the Wear Test M1 (g)	Abrasion Loss Δm (g)	Relative Abrasion Loss Δm/M0 (%)
1#	4.2098	4.1269	0.0829	2.0
2#	4.2928	4.2137	0.0791	1.8
3#	4.2886	4.2093	0.0793	1.8

1#: ZG_70_Cr_2_MnNiSi steel; 2#: Sr-Si-Fe modification; 3#: Re-Ca-Ba modification.

## Data Availability

The data presented in this study are available on request from the corresponding author.

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
