# Peer review of "Investigation of the Thermodynamic Transformation and Rare Earth Microalloying of a Medium Carbon-Medium Alloy Steel for Large Ball Mill Liners"

_materials, 2022, doi:10.3390/ma15134448_

Round 1

Reviewer 1 Report

The following points need to be addressed before the publication of this manuscript.

1.       In the abstract, its better to write “medium carbon-medium alloy steel”.

2.       On Page#2, Lines 77,78, 79, and 80, provide reference for the following text: “ZG70Cr2MnNiSi  steel exhibits the characteristics of wear-resistant steel, which is applicable in the production of lining boards for large ball mills and is also a suitable material for critical components in fitness equipment that require a high bearing capacity.”

3.       In figure 13; the scale bar text is blurr and not easily readable. Improve its resolution to make it readbale.

4.       On Page # 14, Line # 357 (Last line), the degree sign in 218°C is not inserted properly. Make it the same as of page#15, line#361. The same mistake has been repeated at many other occasions in the manuscript. Make the necessary changes accordingly.

5.       On Page#15, Lines 379,380, and 381, provide evidence or reference for the following text: “Due to the high affinity between carbon and the alloy elements in alloy cementite, fine grains of alloy cementite are difficult to dissolve, hindering the aggregation and growth of cementite”.

The conclusions section contains details which are more than necessary. Try to make it concise.

Reviewer 2 Report

The article highlights peculiarities of development of medium carbon medium alloy steel, complex-modified with alkaline earth and rare earth alloys. The authors investigated mechanical properties of the treated steels. They found that rare earth or alkaline earth modification of the medium carbon medium alloy steel promoted microstructural uniformity and grain refinement and improved the mechanical and anti-wear properties. The disadvantage of this article is that the studies presented in Sections 3.1 and 3.2 would be good for steels modified by two rare earth modification agents.

The following corrections must be made:

 1.     The sentence from the “Introduction” section (Lines 82-87) should be moved to the “Materials and Methods” section.

2.     It must be stated from which material a counter body for  the wear test was made.

3.     The wavelength of X-ray radiation should be specified.

4.     The sentence from the text of the manuscript (Line 168-173) should be moved to the caption to Fig. 2.

5.     The sentence from Section 3.2 (Line 207-210) duplicates the sentence from Section 2.3 (Lines 130-134).

6.     The word “sortenite” (Line 233) should be replaced by “sorbite”.

7.     It should be explained for what reasons the heat treatment modes given in the text of the article (Lines 254-269) were chosen.

8.     The sentence from Section 3.3 (Lines 263-265) duplicates the sentence from Іection 2.1 (Lines 90-95).

9.     The authors note an increase in peak intensity for a-Fe (about 30°). However, the presence of reflexes in this range of angles (about 30°) is not characteristic of the crystal structure of a-Fe. Therefore, the authors should check the results and indicate which crystal structure this peak belongs to.

10.  Section 3.3.3 and Section 3.3.4 should be merged, or the authors should suggest another name for Section 3.3.4.

11.  It should be explained why in Section 3.3.4 the authors analyze only ZG70Cr2MnNiSi steel after modification by Sr-Si-Fe.

12.  The sentence from the Section 3.3.5 (Lines 329-330) should be moved to the “Materials and Methods” section.

13.  Wear surface morphology of tested steels should be given at the same magnifications.

14.  “Conclusions” should be shortened. They should be clear and concise.

Round 2

Reviewer 2 Report

The authors took into account all comments of the reviewer and made appropriate corrections to the manuscript.